# MRI Apparent Diffusion Coefficient (ADC) as a Biomarker of Tumour Response: Imaging-Pathology Correlation in Patients with Hepatic Metastases from Colorectal Cancer (EORTC 1423)

**DOI:** 10.3390/cancers15143580

**Published:** 2023-07-12

**Authors:** Alan Jackson, Ryan Pathak, Nandita M. deSouza, Yan Liu, Bart K. M. Jacobs, Saskia Litiere, Maria Urbanowicz-Nijaki, Catherine Julie, Arturo Chiti, Jens Theysohn, Juan R. Ayuso, Sigrid Stroobants, John C. Waterton

**Affiliations:** 1Centre for Imaging Sciences, University of Manchester, Manchester M20 4GJ, UK; alan.jackson@manchester.ac.uk (A.J.);; 2CRUK Cancer Imaging Centre, The Institute of Cancer Research and Royal Marsden NHS Foundation Trust, Downs Road, London SW7 3RP, UK; 3European Organisation for Research and Treatment of Cancer, 1200 Brussels, Belgium; 4EA 4340 BECCOH, UVSQ, Universite Paris-Saclay, 92104 Boulogne-Billancourt, France; 5Department of Pathology, APHP-Hopital Ambroise Pare, 92100 Boulogne-Billancourt, France; 6Nuclear Medicine Unit, IRCCS Humanitas Research Hospital, 20089 Rozzano, Italy; 7Department of Bio-Medical Sciences, Humanitas University, 20072 Milan, Italy; 8Institute of Diagnostic and Interventional Radiology and Neuroradiology, University Hospital Essen, University Duisburg-Essen, 45122 Essen, Germany; 9Radiology Department—CDI, Hospital Clinic Universitari de Barcelona, 08036 Barcelona, Spain; 10Molecular Imaging and Radiology, University of Antwerp, 2000 Antwerp, Belgium

**Keywords:** diffusion-weighted MRI, multicentre trials, quality assurance, quantitation, standardisation

## Abstract

**Simple Summary:**

We hypothesised that change in a magnetic resonance imaging (MRI) biomarker, the apparent diffusion coefficient (ADC) after 14 days of treatment could be a proxy for tumour regression grade (TRG) on pathology. Measurement of the imaging biomarker was standardised across centres. We restricted measurements to liver metastases from colorectal cancer and ensured a standardised chemotherapy approach. We identified and eliminated significant measurement error due to MRI scanner performance. We excluded studies that failed to conform to the imaging protocol or where images contained movement artefact. We ensured stability of the scanners by periodic quality control testing and used a standard, widely used data analysis technique to derive the ADC. Despite these measures, our results showed no significant correlation between ADC and TRG or between ADC and percentage of viable tumour, percentage necrosis, percentage fibrosis or a tumour proliferation index. This may reflect the complex cellular architecture of tumours after treatment.

**Abstract:**

**Background:** Tumour apparent diffusion coefficient (ADC) from diffusion-weighted magnetic resonance imaging (MRI) is a putative pharmacodynamic/response biomarker but the relationship between drug-induced effects on the ADC and on the underlying pathology has not been adequately defined. **Hypothesis:** Changes in ADC during early chemotherapy reflect underlying histological markers of tumour response as measured by tumour regression grade (TRG). **Methods:** Twenty-six patients were enrolled in the study. Baseline, 14 days, and pre-surgery MRI were performed per study protocol. Surgical resection was performed in 23 of the enrolled patients; imaging-pathological correlation was obtained from 39 lesions from 21 patients. **Results:** There was no evidence of correlation between TRG and ADC changes at day 14 (study primary endpoint), and no significant correlation with other ADC metrics. In scans acquired one week prior to surgery, there was no significant correlation between ADC metrics and percentage of viable tumour, percentage necrosis, percentage fibrosis, or Ki67 index. **Conclusions:** Our hypothesis was not supported by the data. The lack of meaningful correlation between change in ADC and TRG is a robust finding which is not explained by variability or small sample size. Change in ADC is not a proxy for TRG in metastatic colorectal cancer.

## 1. Introduction

Diffusion-weighted magnetic resonance imaging (DW-MRI) provides quantitative biomarkers that are used to probe tumour microstructure based on restrictions to water proton diffusion from intra- and extra-cellular structures such as cell membranes and collagen fibres. Commonly used DW-MR protocols in the body allow derivation of an apparent diffusion coefficient (ADC), which reflects water movement over typically 10–40 µm during 10–100 ms; this predominantly represents diffusion in the extravascular extracellular space or in local capillary blood flow. Thus, ADC can be used as an indicator of pathogenic processes, or response to therapeutic interventions. As a response biomarker, ADC could potentially indicate tumour cell death following radiotherapy, oncolytic or cytotoxic therapy as increases in ADC result from loss of cell membrane integrity [1].

The effectiveness of any predictive or response biomarker depends on several factors. First, the repeatability of the biomarker must be established in order to understand the magnitude of change that must be observed to be certain that it represents a true biological effect. Second, it is essential to understand, and compensate for, aspects of biomarker estimation which may cause variations in biomarker measurements and biomarker reproducibility. Third, it is important to confirm that the putative biomarker truly reflects the biological/physiological process which it is believed to demonstrate [2]. The validation of an imaging response biomarker is a complex exercise, but one important approach is to establish how well drug-induced changes in the imaging biomarker reflect drug-induced changes in the underlying pathology.

A number of small prospective studies describe the use of ADC in assessing response of colorectal liver metastases compared to standard Response Evaluation Criteria In Solid Tumours (RECIST) criteria. These studies failed to detect correlation between changes in ADC and lesion size change after 14 days of treatment [3], and noted that ADC change showed an early increase with subsequent decrease, 3 months after treatment [4]. Most importantly, average changes in responding lesions were small compared with the measurement error, suggesting that the techniques as implemented were not reliable enough to predict therapy response at the individual lesion and patient level [5]. To address these challenges, the present study was designed to provide biological validation of ADC changes due to systemic treatment as a response biomarker in liver metastases from colorectal cancer and to test the hypothesis that this could be achieved in a multicentre setting with a prospective trial design using simple ADC metrics.

Multicentre assessment of ADC has previously been challenging. Variations in imaging sequence implementation across scanning platforms, modified acquisition techniques designed to compensate for physiological motion, and variation in analysis techniques [6,7] affect the stability and reproducibility of ADC measurements. Therefore, to mitigate these risks, we designed a prospective study that employed validated data acquisition protocols and standardized analyses within a strong quality assurance and quality control framework [8]. We tested our hypothesis that early drug-induced changes in ADC faithfully reflect biological response in patients with hepatic metastases from colorectal cancer.

## 2. Materials and Methods

### 2.1. Trial Design

This was a prospective, multicentre, single-arm imaging trial listed on clinical trials.gov (NCT02355353). All patients gave their informed consent for inclusion before they participated in the study. The study was conducted in accordance with the Declaration of Helsinki, and the protocol was approved by the ethics committees of the individual institutions (EORTC 1423). Patients with histologically confirmed clinical stage IV metastatic colorectal cancer (mCRC) with metachronous or synchronous liver metastases who were candidates for neoadjuvant therapy were eligible. Three pre-operative treatment regimens were considered acceptable to facilitate recruitment: chemotherapy alone, chemotherapy + anti-angiogenic agent, or chemotherapy + monoclonal antibody. Twenty-six patients (14 male, 12 female) aged 45–82 years (mean 64.85 ± 9.12 years) were recruited, gave written, informed consent, and started the study treatment.

The design of the study is shown in Figure 1. The enrolled patients received up to 6 cycles preoperative therapy based on their treating physicians’ best choice. DW-MRI was performed at baseline and 14 days after the start of therapy. Repeat imaging was performed within one week prior to subsequent surgery.

Metastatic lesions were identified on the baseline MR scan. All individual metastases, greater than 1 cm in diameter were identified and the hepatic lobe from which the lesion was thought to arise was noted. Each lesion was marked on the original images and given a unique identification number. These annotated images were subsequently used for lesion identification by both the image analysis and surgical teams.

The primary objective was the correlation between the percentage of ADC change at day 14 relative to baseline and tumour regression grade (TRG) in the surgical resection specimen.

Secondary objectives were the following:To measure the variability of test–retest ADC measurements at baselineTo correlate pre-operative (post-treatment) ADC measurement and TRGTo correlate pre-operative (post-treatment) ADC measurement and tumour tissue cellularity/density, necrosis, and proliferation (Ki-67)

### 2.2. Image Handling

#### 2.2.1. Quality Control (QC)

Scanner QC tests were performed at each centre before the centre was allowed to recruit. A diffusion phantom was constructed and supplied to each centre [9]; details on phantom acquisition and QC procedures are included in the Appendix A.

Centres and scanner types included are presented in Table 1.

The QC guidelines were based on pooled group performance for each metric on the initial QC [9] scan from each site. Any scanner falling outside the quality threshold (mean ± 1 SD) for any of the QC metrics was rejected and the initial phantom scan repeated (Appendix A). Phantom QC scans were then performed on a 6-monthly basis for all centres that passed initial QC assessment.

#### 2.2.2. Scanning Protocol

The central scanning protocol consisted of standardized T1- and T2-weighted axial images for anatomical localization together with a geometrically matched diffusion weighted imaging sequence. Image acquisition protocols are shown in Table 2. Additional sequences as required by the local clinical team for assessment or surgical planning were permitted.

No changes were allowed in field-of-view, pixel size, acquisition matrix, b-values, number of slices, fat suppression technique, diffusion gradient mode, slice gap, or number of signal acquisitions. Scans which demonstrated any such deviations were excluded from the study.

#### 2.2.3. Measurement of ADC

Parametric maps of ADC were calculated voxelwise for the whole data volume using a mono-exponential fit model implemented in the ADC plugin in Osirix MD [10]. All metastatic lesions identified on the pre-treatment scan were delineated at each time point to produce tumour specific volumes of interest (VOI). Manual tumour delineation was undertaken by a single experienced radiologist (AJ). Delineation was performed primarily on b = 1000 DW images but with access to all other imaging sequences. A randomly selected group of 25 lesions was redefined by the same observer with an interval of 4–6 weeks in order to determine reproducibility of the VOI definitions and by a second blinded observer (RP) to determine inter-observer variation. Histogram analysis of repeatability and limits of agreement for individual tumours, were also calculated using a 5% level of significance. Within-patient coefficient of variance (CoV) for repeated measures was used to compared baseline repeatability for mean ADC. ΔADC% between test-retest ADC was calculated as a measure of repeatability for individuals. The 95% confidence interval width (95%CIw); defined as the width of the 95% confidence bound between the lower confidence interval, or higher confidence interval, and the mean, was calculated for each measurement by both observers. From each VOI a selection of variables was estimated: (1) tumour volume; (2) mean ADC (ADCmean); (3) maximum ADC (ADCmax). The percentage change in tumour volume (TV) and in mean and maximum ADC were calculated between baseline and day 14 (∆TV_early_ and ∆ADC_early_ respectively) and between baseline and the pre-operative scan (∆TV_late_ and ∆ADC_late_ respectively).

### 2.3. Histological Evaluation

At surgery, tissue was collected from all resected lesions. The lesion was identified according to the code attributed on the original pre-surgical scans. Haematoxylin and eosin (H&E) stained slides were used to provide data on tissue cellularity.

Pathological response was independently determined on a lesion-by-lesion basis at a central reference laboratory. This was carried out independently by two experienced histopathologists (CJ and MU-N) and discrepancies reconciled by consensus discussion and agreement. The pathological response rate was graded for five tumour response groups using the criteria established by Rubbia-Brandt [11]. The proposed scoring system is based on the presence of residual tumour cells and the extent of fibrosis (TRG1 = absence of tumour cells replaced by abundant fibrosis; TRG2 = rare residual tumour cells scattered throughout abundant fibrosis; TRG3 = residual tumour cells throughout a predominant fibrosis; TRG4 = large number of tumour cells predominating over fibrosis; TRG5 = almost exclusively tumour cells without fibrosis) [12].In addition, both pathologists similarly estimated and agreed on viable tumour cell surface area and expressed it as percent of total surface area of the metastatic lesion. Total surface area included tumour, stroma, immune infiltrates, and necrotic areas. Tumour necrosis was quantified as follows:necrotic area/(necrotic area + non-necrotic solid tumour)

Analysis of Ki-67 IHC stained slides provided data on tumour proliferation. The Ki-67 status was expressed as the percentage of Ki-67 positive nuclei to all tumour nuclei.

### 2.4. Statistical Analysis

Co-primary endpoints for imaging were ΔADC_early_ defined on ADCmean and ADCmax. The primary endpoint measures for pathology were TRG by Rubbia-Brandt criteria. Percentage of viable tumour and necrosis, proliferative activity (Ki-67 index), and changes in tumour volume were secondary endpoints. In order to demonstrate with 95% confidence interval (one sided) that the correlation between the imaging biomarker change and the pathological response is <−0.5 (H0: rho ≥ −0.5) with 90% power if the true correlation is −0.8 (H1: rho < −0.5), 31 lesions were needed.

The relationship between imaging parameters (ADCmean and ADCmax) and histological features (percentage viable tumour, percentage fibrosis, percentage necrosis, and Ki-67 index) were examined using Spearman rank correlation with Fisher transformation. The relationship between baseline (mean ADC, day 14 ADC, ∆ADC_early_, and ∆ADC_late_) and TRG was also examined using Spearman rank correlation. A sensitivity analysis to include the dependency of lesions from the same patient was performed with a repeated measures model using the standardized ranked variables. The one-sided type I error was fixed at 1% for all secondary and exploratory analyses. When relevant, equivalent two-sided 98% confidence intervals were reported. The statistical package used was SAS 9.4 (STAT 14.3).

## 3. Results

### 3.1. Quality Control (QC)

The majority of the scanners (13/16) showed less than 5% deviation through the phantom quality control process (details in Appendix A). After error corrections, one scanner was finally omitted from the study due to QC failure. Once clinical scanning commenced, initial QC assessment showed excellent conformance to the scan protocols although significant movement artefact was noted in four scans.

Single-rater reproducibility of tumour volume showed a mean difference in percentage tumour volume of (0.4 ± 7.9)% with 95% confidence limits of ±13.2%. This variation was driven largely by two very small lesions with high reproducibility errors: when these were removed the 95% confidence limit width became ±7.4%. Inter-rater reproducibility showed a mean difference in tumour volume of (0.48 ± 8.3)% with 95% confidence interval ±14.6%. When the two small lesions were removed this became ±8.2%. The group CoV between test and retest mean ADC for observer A was 8.8% ± 3.4%. For observer B, the CoV for mean ADC was 7.6% ± 2.1%.

### 3.2. Patient Demographics

Patient recruitment occurred at 8 of the 10 centres. A total of 29 patients were registered for the study of which two were ineligible. One further patient declined chemotherapy and 26 were therefore entered into the trial. Eighteen of the 26 patients received chemotherapy alone, seven received a combination of chemotherapy plus another drug (bevacizumab [n = 5], or panitumumab [n = 2]) and one patient commenced chemotherapy but panitumumab was added after one cycle. Statistical analysis was performed on all patients together and separately on the chemotherapy only cohort.

The patient group consisted of 14 men and 12 women with a median age of 64.5 years (range: 45–82 years). The chemotherapy-only patient cohort consisted of 10 men and 8 women, with a median age of 64.5 years (range: 45–82 years). The primary diagnosis was colon cancer in 16/26 and rectal cancer in 10/26 cases. The histological grade and TNM staging at first diagnosis are shown in Table 3.

Surgical resection was eventually performed in 23 patients (16 had received chemotherapy alone). The majority of the patients had more than one lesion resected: however, for one patient the day 14 DW-MRI was not available and for another patient no surgical specimen was available for histopathology. Therefore, in total, there were 46 evaluable lesions from 21 patients for the primary objective.

### 3.3. Imaging Findings

Baseline scanning included 82 lesions. The number of measured lesions decreased with time. Tumour volume ranged from 0.3–501.2 cm^3^ at baseline (median 8.4 cm^3^, mean 34.55 cm^3^, SD 77.22 cm^3^), was unchanged after 14 days ranging from 0.3–506.1 cm^3^ (median 7.6 cm^3^, mean 33.34 cm^3^, SD 83.76 cm^3^), and decreased prior to surgery ranging from 0–254.1 cm^3^ (median 6.5 cm^3^, mean 22.8 cm^3^, SD 45.0 cm^3^). Tumour volume distribution was skewed towards the right, principally by a single extremely large lesion.

Baseline measurements of mean and maximum ADC are shown in Table 4 for all imaged patients and separately for those proceeding to metastatectomy. In the latter group, three small lesions were no longer visible after 14 days of chemotherapy and could not therefore be measured. A further five lesions were no longer visible one week prior to surgery and in one case, two lesions had merged to form a single lesion.

Baseline ADCmean had a median value of 1.0 × 10^−3^ mm^2^/s with a range of 0.7–2.7 × 10^−3^ mm^2^/s. The ADCmax and ADCmean increased for some patients between baseline and day 14, while for others it decreased. Overall, for all patients, ADCmean increased at day 14 with ∆ADC_early_ of 4.3% and a range of −38.3% to 94.8%. In the chemotherapy-only cohort ADCmean increased at day 14 with ∆ADC_early_ of 4.9% and a range of −26.5 to 94.8%. ADC values at day 14 showed no significant difference from baseline (*p* = 0.94). Values of ∆ADCmean_early_ and ∆ADCmean_late_ are shown in Table 5.

For patients undergoing surgery (n = 23), values of ∆ADCmean_early_ and ∆ADCmean_late_ are shown in Table 6.

### 3.4. Surgical Specimens and Pathology

Twenty-seven of 48 lesions were graded as TRG4, 7 as TRG3,12 as TRG2 and 1 as TRG1. In one case the lesion was too small to grade. There was no difference between TRG in the chemotherapy-only and combination therapy groups. There was very wide variation in lesion surface area, fibrosis, necrosis, and the total surface area of viable tumour cells (Table 7).

### 3.5. Correlation between Imaging Biomarkers, Tumour Response Grade, and Histology

There was no evidence of significant correlation (*p* = 1.0) between TRG and ∆ADCmean_early_ (study primary objective). In scans acquired one week prior to surgery, there was no significant correlation between ADCmax or ADCmean and percentage of viable tumour, percentage necrosis, percentage fibrosis, or Ki-67 index (Table 8).

## 4. Discussion

Our hypothesis that an increase in ADC after 14 days of neoadjuvant chemotherapy and measured immediately before tumour resection would reflect underlying histological features of tumour response as measured by TRG was not supported by the data. At a pathophysiological level, a relationship between ADC and percentage tumour necrosis is to be expected. As free water molecular motion quantified by the ADC is affected by the structure of the tumour microenvironment, it is expected that in response to therapy, ensuing cell death will lead to loss of cellular integrity with consequent increase in diffusion. ADC therefore has been widely proposed as an early response biomarker for chemo- and radiotherapy [13,14]. Our findings, despite meticulous quality control and careful study design, were disappointing and can be interpreted in two ways. First, it may be that no such relationship exists or, second, it may be that an underlying biological relationship does exist but that confounding factors, such as the complexity and variability of tumour structure within hepatic metastases following therapy, masks any correlation. This has been shown to be the case in ovarian cancer where the ADC increases differentially at metastatic disease sites despite similar tumour shrinkage; the negative correlation of post-chemotherapy ADC with tumour cell fraction and positive correlation of change in ADC with percentage necrosis was present for peritoneal but not omental or lymph node metastases [15]. A small single centre study in mCRC which showed good ADC repeatability indicated that ADC values from untreated liver metastases were correlated with the proliferative marker Ki-67 at histopathology of resected specimens [16]. A larger single centre retrospective study showed that ADC values from whole tumour as well as peripheral ring and central core taken post chemotherapy and immediately prior to resection were negatively correlated with the TRG classes [17]. This has been corroborated by other studies where the post-chemotherapy ADC and ΔADC values were significantly different between TRG groups [18] or between responders and non-responders [19]. In contrast, a prospective study that looked at the change in ADC with neoadjuvant treatment and prior to surgery showed increases in both responding (TRG 1–2) and non-responding (TRG 3) patients [20]. However, low baseline and post-treatment ADC in liver metastases from mCRC were purported to have better outcomes but were unrelated to histology in another series [21]. Nevertheless, despite the uncertain relationship of ADC with TRG, the utility of DWI for assessment of response to chemotherapy is supported by a systematic review of 16 studies, nine of which utilised MRI [22].

The stability and repeatability of ADC measurements in this multicentre setting, which critically affects the performance of a biomarker [2], should have been adequate to test our hypothesis. The baseline reproducibility of ADC metrics on three of the scanners from different vendors in this study investigated in a preliminary study of 20 patients with colorectal liver metastases and imaged using the same protocols as in the current study showed a coefficient of variation (CoV) of 6.2% and 95% limits of agreement of 21% [7]. This is highly comparable with the results from other single centre studies; the CoV of ADC measurements in abdominal organs in single centre studies has been shown to vary between 1.7 and 7% [7,23] with 95% limits of agreement ranging from approximately 10–30%. In comparison, in multicentre studies, ADC measurement variability is greater as a result of differences between imaging platform, manufacturer, scanner model, and field strength [6,24]. This was demonstrated in normal healthy volunteers using different scanners and was particularly notable in the liver [6,25] but was also seen in other abdominal organs including the kidney and pancreas, where it was most marked at 3 T. A multicentre cross vendor study from the Cancer Core Europe task force indicated that in healthy volunteers, the measurement variation for liver ADC in two repeated scans did not exceed 11% while measurement variation between sites amounted to 20% [26]. Other factors that significantly affect reproducibility include the stability of the imaging system [27] as well as the structure of the tumour itself [28] and the curve fitting technique used in the analysis [13,29]. In the current study we standardised imaging protocols to the greatest degree possible between vendors and field strengths. We also ensured that analysis was performed at a single central site using a standard analysis package and that we had previously established the repeatability of ADC measurements on a subgroup of the multivendor machines [7]. We identified and eliminated significant measurement error due to scanner performance and failure to conform to the imaging protocol or physiological movement. We ensured stability of the imaging systems by periodic quality control testing, we standardised the tumour type and therapeutic approach, and we utilised a standard mono-exponential curve fitting technique which is widely used in diagnostic diffusion imaging. In general, the accuracy of ADC measurements from the phantom test object across sites and field strengths was good and known ADC measurement differences were due to errors in the use or setup of the phantom itself. A single scanner was excluded from the study due to inaccurate ADC measurements. Although the mean ADC changes in the patient cohort after treatment were 4.3% and 4.9%, the actual range of changes was −38.3 to 94.8% and −26.5 to 94.8%. This makes an estimated measurement error of 4 to 5% quite acceptable for the detection of changes of this size, as measurement reproducibility is useful when applied to individual subjects not to a group as a whole. However, small changes in ADC in individual patients would require a more advanced analysis with motion correction and tumour size modelling, which is a limitation of this study.

Other factors which can affect repeatability and contribute to measurement error are the lesion segmentation process and the tumour volume being segmented. The 2D methods result in more variability of the derived biomarker because of slice selection bias [30], whereas 3D volumes, as used in the current study, are considered to provide more accurate and reproducible estimations [31]. However, even small errors in ROI delineation can affect mean/median tumour values [32] and ROI placement is important for ensuring repeatability [33]. In the current study we found a mean difference in tumour volume and test-retest of 0.4% with a single rater with 95% confidence limits of only 7.4% when two very small lesions were removed from the analysis. In addition, following removal of these two very small lesions from the test set, inter-rater reproducibility was also good, with a mean difference in tumour volume of 0.48% and a confidence limit of 8.2%. The effect of tumour volume on ADC reproducibility was demonstrated in a multi-study meta-analysis [23] where high repeatability (CoV 1.7%) was found in one study, typified by large tumour volumes. This study was in fact an outlier and so was removed prior to calculation of group repeatability values. Statistical variations in the estimation of mean values contribute primarily to the variations in repeatability; estimation of the mean value is much less accurate when very few samples are available for its calculation, thus small tumours with few pixels will be subject to significant estimation error [34]. A recent study described a statistical error model which can be used to correct for these estimation errors. Use of the model in a multicentre/multivendor study of hepatic colorectal metastases showed a reduction in the threshold for identifying biological response from 21 to 2.7%. In a small single- centre study of liver metastases from colorectal cancer, where both motion and statistical estimation error were corrected, the threshold for identifying a biological response fell dramatically from 30% to 1.2% [7]. In fact, changes in ADC within seven days of starting treatment were predictive of tumour volume response and showed close agreement with changes in lactate dehydrogenase level which is currently used at an early response biomarker in some clinical applications.

We made no attempt in the current study to compensate for the effects of physical motion on the estimation of ADC except for excluding patients with subjective evidence of image degradation in the quality control review. As ADC is estimated by curve fitting using single pixel values of images acquired with varying strengths of gradient, physiological motion during the acquisition can cause significant misregistration between images and can introduce error into estimates of ADC respiratory triggering or the use of navigator echo techniques can mitigate motion effects when compared to free-breathing studies [35]. Similarly, a post-processing co-registration technique can produce improvements in ADC reproducibility of up to 6-fold depending on the amount of physiological motion present in the initial image [36]. In a recent single centre study examining the effects of chemotherapy on colorectal liver metastases, we examined the effects of physiological motion and statistical sampling error on the repeatability of ADC metrics [7]. Patients with clear subjective physiological motion artefact were excluded as they were in the current study. Motion correction reduced the CoV between test-retest images from 9.8 to 3.2% and the threshold for identifying a biological response from 30% to <9%.

Our study has limitations: the sample size was small despite the prevalence of liver metastases in colorectal disease. This was because of the need to specifically include patients suitable for metastatectomy who consented to undergoing neoadjuvant chemotherapy. The strict criteria which also excluded the use of certain neoadjuvant agents such as anti-angiogenic agents and monoclonal antibodies because of potential confounding effects on pathological response [37] also negatively impacted recruitment. Pathological assessment was by experienced observer assessment rather than using newer more objective digital pathology techniques [38]. Use of digitised histopathological objective assessment of whole lesion necrosis and cell density would be advantageous in future analyses. There was little intra-patient correlation from the presence of multiple lesions per patient and this may have introduced bias in our analysis. Finally, we cannot exclude the alternate interpretation that ADC-measurable tumour microenvironment changes do happen in response to treatment, but not necessarily at the pre-specified 14-day interval that was analysed.

## 5. Conclusions

We demonstrated that it is feasible to conduct a rigorous, quality controlled multicentre study using ADC within a clinical trial setting; when applied in clinical practice this should enable personalised therapy. Nevertheless, we did not demonstrate any significant relationship between early changes in ADC induced by therapy and TRG and we found there is no relationship between tumour ADC measured in the post-therapy setting and common histological features. We believe that this is likely to reflect, at least in part, the complex structure of the tumour micro-environment following therapy.

## Figures and Tables

**Figure 1 cancers-15-03580-f001:**
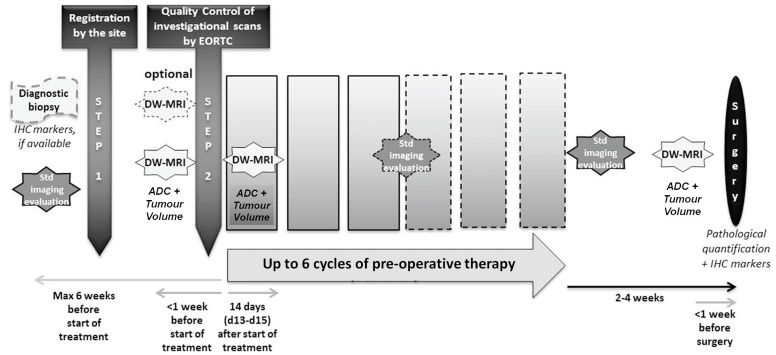
Diagram showing trial design.

**Table 1 cancers-15-03580-t001:** Centres, scanners, and number of cases scanned.

Institution	Patients	MRI Scanners
Humanitas Unversity (Milan, Italy)	10	Phillips Achieva 1.5 T, Phillips Ingenia 1.5 T
Institut Bergonié (Bordeaux, France)	5	Siemens Aera 1.5 T
Universität Duisberg-Essen (Essen, Germany)	4	Siemens Aera 1.5 T
Universitari de Barcelona (Barcelona, Spain)	3	Siemens Aera 1.5 T
Hospital Universitari La Fe (Valencia, Spain)	2	GE Optima 360 3 T
Medizinische Universität Wien (Vienna, Austria)	2	Siemens Trio 3.0 T
Sapienza Università di Roma (Rome, Italy)	2	Siemens Avanto 1.5 T
Charité—Universitätsmedizin (Berlin, Germany)	1	Siemens Aera 1.5 T

**Table 2 cancers-15-03580-t002:** Scanning parameters for T1 and T2-weighted and diffusion-weighted imaging at 1.5 T and 3 T. * The parallel imaging scheme used for diffusion-weighted imaging on 1.5 T and 3 T Siemens scanners was GRAPPA, for GE scanners was ASSET, and for Philips scanners was SENSE.

Parameter	T1 and T2 Anatomical	Diffusion Weighted 1.5 T	Diffusion Weighted 3 T
FOV	380	380	380
Pixel Size	1.5 × 1.5 mm	3 × 3 mm	3 × 3 mm
Slice Thickness	5 mm	5 mm	5 mm
Slice gap	0	0	0
Respiratory Control	Breath holding if required	Free breathing	Free breathing
Acquisition matrix	256 × 224 (87.5%)	128 × 112 (87.5%)	240 × 240 (87.5%)
Reconstruction matrix	256 × 256	256 × 256	256 × 256
Number of slices	40	40	40
Number of signal averages	2	4	4
TR ms	Site specific	≥8000	≥5000
TE ms	Site specific	minimum	minimum
Parallel imaging *	Site specific	yes	yes
Acceleration factor	Not specified	2	2
Diffusion gradient mode	Not applicable	3 scan-trace	3 scan-trace
Fat suppression	None	SPAIR	SPAIR
b-values s/mm^2^	Not applicable	100, 400, 800	150, 400, 800

**Table 3 cancers-15-03580-t003:** Primary diagnosis, histological grade, and TNM staging at diagnosis for the whole cohort and the chemotherapy alone group.

	All Patients (N = 26)	Chemotherapy Alone (N = 18)
	N (%)	N (%)
Site of the primary tumor		
Colon cancer	16 (61.5)	12 (66.7)
Rectum cancer	10 (38.5)	6 (33.3)
Histological grade		
GI	3 (11.5)	2 (11.1)
GII	12 (46.2)	7 (38.9)
GIII	6 (23.1)	4 (22.2)
Missing	5 (19.2)	5 (27.8)
TNM staging at first diagnosis		
Stage I	2 (7.7)	0 (0.0)
Stage IIA	4 (15.4)	3 (16.7)
Stage IIB	2 (7.7)	1 (5.6)
Stage IIIB	1 (3.8)	1 (5.6)
Stage IVA	16 (61.5)	12 (66.7)
Stage IVB	1 (3.8)	1 (5.6)

**Table 4 cancers-15-03580-t004:** Median and Mean ADCmax and ADCmean values at baseline, Day 14 (ADC_early_) and prior to surgery (ADC_late_) in evaluable lesions in 26 imaged patients at these 3 time points and in the surgical cohort alone. Surgery was performed in 23 patients who had 48 lesions in the liver at presentation. By day 14, 3 of these had responded to chemotherapy, and a further 6 resolved immediately prior to surgery. Surgical specimens were not available in 1 patient.

		All Patients			Patients Undergoing Surgery		
		Baseline	Day 14	Within 1 Week before Surgery	Baseline	Day 14	Within 1 Week before Surgery
ADC max (10^−3^ mm^2^/s)	No. lesions measured	82	79	65	48	46	39
	Median	2.0	2.1	2.0	2.1	2.2	1.9
	Range	1.1–4.1	1.1–4.1	1.3–4.1	1.1–4.1	1.1–4.1	1.3–4.1
	Mean (SD)	2.21 (0.73)	2.23 (0.70)	2.21 (0.72)	2.24 (0.65)	2.21 (0.61)	2.08 (0.60)
ADC mean (10^−3^ mm^2^/s)	No. lesions measured	82	79	65	48	46	39
	Median	1.0	1.1	1.1	1.0	1.1	1.1
	Range	0.7–2.7	0.7–2.6	0.4–2.6	0.7–2.7	0.7–2.6	0.8–2.5
	Mean (SD)	1.19 (0.45)	1.24 (0.41)	1.29 (0.45)	1.16 (0.39)	1.19 (0.36)	1.20 (0.36)

**Table 5 cancers-15-03580-t005:** Median and mean values of the change in ADCmax and ADCmean at Day 14 (ΔADC_early_) and immediately prior to surgery (ΔADC_late_) in evaluable lesions in all patients and in those receiving chemotherapy alone imaged at 3 time points. The ADCmean showed a small increase but the ADCmax did not.

		ΔADCmax		ΔADCmean	
		All Patients N = 26	Chemotherapy Alone N = 18	All Patients N = 26	Chemotherapy Alone N = 18
ΔADC_early_ (%)	No. lesions measured	79	56	79	56
	Median	−0.7	0.9	4.3	4.9
	Range	−28.6–60.5	−28.6–40.4	−38.3–94.8	−26.5–94.8
	Mean (SD)	1.90 (16.27)	1.94 (15.29)	5.43 (17.02)	7.35 (16.51)
ΔADC_late_ (%)	No. lesions measured	65	48	65	48
	Median	−1.5	−1.2	7.6	6.4
	Range	−50.8–61.8	−50.8–61.8	−64.1–116.2	−64.1–116.2
	Mean (SD)	−0.23 (21.28)	0.41 (20.80)	9.90 (23.65)	9.07 (25.38)

**Table 6 cancers-15-03580-t006:** Median and mean values of the change in ADCmax and ADCmean at Day 14 (ΔADC_early_) and immediately prior to surgery (ΔADC_late_) in evaluable lesions in 23 patients (16 who had chemotherapy) imaged at 3 time points who subsequently underwent metastatectomy.

		ΔADCmax		ΔADCmean	
		All Surgical Patients (N = 23)	Chemotherapy Alone (N = 16)	All Surgical Patients (N = 23)	Chemotherapy Alone (N = 16)
ΔADC_early_ (%)	No. lesions measured	46	33	46	33
	Median	−1.4	−2.8	5.2	5.5
	Range	−28.6–40.4	−28.6–40.4	−38.3–37.5	−26.5–37.5
	Mean (SD)	−0.70 (15.37)	−0.93 (15.59)	2.82 (14.84)	4.98 (11.65)
ΔADC_late_ (%)	No. lesions measured	39	29	39	29
	Median	−6.8	−6.8	4.0	2.2
	Range	−50.8–61.8	−50.8–61.8	−18.6–54.5	−16.1–44.8
	Mean (SD)	−6.01 (21.44)	−6.76 (20.52)	7.58 (18.71)	6.01 (18.18)

**Table 7 cancers-15-03580-t007:** Quantitative histological evaluation of tumour components.

	Total Surface Area of Lesion (mm^2^)	Total Surface Area of Fibrosis (%)	Total Surface Area of Necrosis (%)	Total Surface Area of Viable Tumour Cells (%)	Ratio Ki-67 Positive to Total Tumour
Median	180.0	37.5	20.0	32.1	0.2
Range	3.0–813.0	5.0–95.8	0.0–82.9	0.0–80.0	0.0–0.6
Mean (SD)	227.35 (195.29)	42.42 (24.19)	23.64 (21.37)	33.94 (22.02)	0.22 (0.17)

**Table 8 cancers-15-03580-t008:** Correlation between imaging and histopathological metrics with significance levels.

		All Patients		Chemotherapy Only	
		ADCmax (n = 39)	ADCmean (n = 39)	ADCmax (n = 29)	ADCmean (n = 29)
% viable tumour	Correlation	−0.404	−0.222	−0.411	−0.280
	*p*-value	0.776	0.976	0.732	0.916
% necrosis	Correlation	0.005	−0.384	−0.132	−0.632
	*p*-value	1.00	1.00	1.00	1.00
% fibrosis	Correlation	0.213	0.386	0.311	0.646
	*p*-value	0.979	0.815	0.886	0.142
Ki-67	Correlation	−0.115	−0.329	−0.238	−0.459
	*p*-value	0.996	0.901	0.946	0.625

## Data Availability

The trial data is available for future research as per the EORTC data sharing policy https://www.eortc.org/policies-guidelines/.

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
