# Peer review of "MRI Apparent Diffusion Coefficient (ADC) as a Biomarker of Tumour Response: Imaging-Pathology Correlation in Patients with Hepatic Metastases from Colorectal Cancer (EORTC 1423)"

_cancers, 2023, doi:10.3390/cancers15143580_

Round 1

Reviewer 1 Report

This paper described a multi-center study aiming to investigate the correlation between early and late ADC changes relative to baseline with pathological treatment response indicators for patients with hepatic metastases from colorectal cancer. This work was well designed and aims to answer a very important question even though the study results are inconclusive. I would appreciate the authors to respond to the following comments:

1.        It will be helpful to include a table to list all the centers and main specifications for the MRI scanners being used at these centers.

2.        The supplementary documents provided a detailed description for the very important QC requirement for participating centers. The QC guidelines include tolerance based on pooled group performance, such as 5% for repeatability variations and 15% for fitting errors. Could you please discuss how this will impact the results given that patient ADC is around the same range, 4.3% and 4.9%, etc?

3.        Could you please share more results in terms of inter-observer variation of VOI definition/ tumor segmentation? Tumor volume by itself is not enough for evaluating segmentation repeatability.

Author Response

Reviewer 1

This paper described a multi-center study aiming to investigate the correlation between early and late ADC changes relative to baseline with pathological treatment response indicators for patients with hepatic metastases from colorectal cancer. This work was well designed and aims to answer a very important question even though the study results are inconclusive.

I would appreciate the authors to respond to the following comments:

  1. It will be helpful to include a table to list all the centers and main specifications for the MRI scanners being used at these centers.

Table 1 has been added in the Methods to include this information and subsequent tables have been renumbered.

Institution

Patients

MRI scanners

Humanitas Unversity (Milan, Italy) 

10 

Phillips Achieva 1.5T, Phillips Ingenia 1.5T 

Institut Bergonié (Bordeaux, France) 

Siemens Aera 1.5T 

Universität Duisberg-Essen (Essen, Germany) 

Siemens Aera 1.5T 

Universitari de Barcelona (Barcelona, Spain) 

Siemens Aera 1.5T 

Hospital Universitari La Fe (Valencia, Spain)

GE Optima 360 3T 

Medizinische Universität Wien (Vienna, Austria) 

Siemens Trio 3.0T 

Sapienza Università di Roma (Rome, Italy) 

Siemens Avanto 1.5T 

Charité – Universitätsmedizin (Berlin, Germany) 

Siemens Aera 1.5T

  1. The supplementary documents provided a detailed description for the very important QC requirement for participating centers. The QC guidelines include tolerance based on pooled group performance, such as 5% for repeatability variations and 15% for fitting errors. Could you please discuss how this will impact the results given that patient ADC is around the same range, 4.3% and 4.9%, etc?

Thank you for this valuable comment. We have added at the end of paragraph 2 in the Discussion “Although the mean ADC changes in the patient cohort after treatment were 4.3% and 4.9%, the actual range of changes was -38.3 to 94.8% and -26.5 to 94.8%.  This makes an estimated measurement error of 4 to 5% quite acceptable for the detection of changes of this size as measurement reproducibility is useful when applied to individual subjects not to a group as a whole.  However, small changes in ADC in individual patients would require a more advanced analysis with motion correction and tumour size modelling, which is a limitation of this study.”

  1. Could you please share more results in terms of inter-observer variation of VOI definition/ tumor segmentation? Tumor volume by itself is not enough for evaluating segmentation repeatability.

Thank you, we have added the following to the Measurement of ADC section of the Methods “Histogram analysis of repeatability and limits of agreement for individual tumors, were also calculated using a 5% level of significance. Within-patient coefficient of variance (CoV) for repeated measures was used to compared baseline repeatability for mean ADC.  ΔADC% between test-retest ADC was calculated as a measure of repeatability for individuals. The 95% confidence interval width (95%CIw); defined as the width of the 95% confidence bound between the lower confidence interval, or higher confidence interval, and the mean, was calculated for each measurement by both observers.”

Also we added the following to the QC section of the Results:

“The group CoV between test and retest mean ADC for observer A was (8.8±3.4) %. For observer B, the CoV for mean ADC was (7.6±2.1) %.”

Reviewer 2 Report

Alan Jackson and coworkers reported MRI apparent diffusion coefficient (ADC) as a biomarker of tumour response: imaging-pathology correlation in patients with

hepatic metastases from colorectal cancer (EORTC 1423). The experimental is carefully conducted, and the results have been presented correctly, and the contents fall well into the scope of the journal. However, some points need to be addressed/answered before further considered:

1. The layout is too messy, especially the pictures and tables, suggest a good revision.

2. Line 82: ...was quantified as should be changed to ... was quantified as:.

3. Do not separate between headers and tables (Table 6).

4. All headings and tables should be in the same order, i.e. headings at the top, tables at the bottom.

5. Why are there so many empty cells between Table 7 and the conclusion?

6. Please cite the following two papers if you can: Micropor. Mesopor. Mater., 2022, 112098; and Inorganics, 2022, 10, 202.

Author Response

Reviewer 2

Alan Jackson and coworkers reported MRI apparent diffusion coefficient (ADC) as a biomarker of tumour response: imaging-pathology correlation in patients with hepatic metastases from colorectal cancer (EORTC 1423). The experimental is carefully conducted, and the results have been presented correctly, and the contents fall well into the scope of the journal. However, some points need to be addressed/answered before further considered:

  1. The layout is too messy, especially the pictures and tables, suggest a good revision.

We have standardised the layout including fonts and line spacing for all figures and tables in compliance with the instructions provided at https://www.mdpi.com/journal/cancers/instructions

  1. Line 82: “...was quantified as” should be changed to “... was quantified as:”.

Change made.

  1. Do not separate between headers and tables (Table 6).

Change made

  1. All headings and tables should be in the same order, i.e. headings at the top, tables at the bottom.

Change made

  1. Why are there so many empty cells between Table 7 and the conclusion?

This appears to be a formatting issue.  We have attempted to reduce the white space in our version of the document, and we hope it now renders correctly for the reviewer.

  1. Please cite the following two papers if you can: “Micropor. Mesopor. Mater., 2022, 112098; and “Inorganics, 2022, 10, 202”.

We do not understand the comment, which seems to refer to papers entitled “A microporous 2D cobalt-based MOF with pyridyl sites and open metal sites for selective adsorption of CO2” and “A 2D Porous Zinc-Organic Framework Platform for Loading of 5-Fluorouracil”, neither of which bears any relationship to our work.  Perhaps the reviewer has inadvertently pasted that text in from a different document?

Reviewer 3 Report

The authors present a study to evaluate the apparent diffusion coefficient obtained from diffusion-weighted imaging MRI as a potential treatment response biomarker in patients with liver metastases from colorectal cancer, by comparing it to the tumor regression grade. The trial was prospectively registered and significant efforts were described to diminish the variability across devices or centers. The ad hoc power analysis that was conducted to determine the minimum number of lesions to be evaluated enables the drawing of conclusions as presented in the manuscript. Rigorous QA and standard of reference methods were employed including phantom imaging and the use of a pathological ground truth. Overall, the arguments are solid and backed by data. I raise minor comments below:

- In Table 1, specify the type(s) of parallel imaging used (GRAPPA, CAIPIRINHA, Multi-Band, Simultaneous Multislice)

- Regarding the single-rater reproducibility of tumor volume, which showed higher variability driven largely by two very small lesions, please clarify how this relates to the minimum 1 cm lesion size for the specified inclusion criteria. Are those exclusively post-treatment lesions that regressed? If so, would they not be the most important to capture ADC changes pertaining to tumor regression, since the lesions were the most responsive to treatment as measured by standard criteria (e.g., RECIST)?

- In the discussion, I would consider the alternate interpretation that ADC-measurable tumor microenvironment changes do happen in response to treatment, but not necessarily at the pre-specified 14-day interval that was analyzed.

Author Response

Reviewer 3

The authors present a study to evaluate the apparent diffusion coefficient obtained from diffusion-weighted imaging MRI as a potential treatment response biomarker in patients with liver metastases from colorectal cancer, by comparing it to the tumor regression grade. The trial was prospectively registered and significant efforts were described to diminish the variability across devices or centers. The ad hoc power analysis that was conducted to determine the minimum number of lesions to be evaluated enables the drawing of conclusions as presented in the manuscript. Rigorous QA and standard of reference methods were employed including phantom imaging and the use of a pathological ground truth. Overall, the arguments are solid and backed by data. I raise minor comments below:

- In Table 1 [now Table 2], specify the type(s) of parallel imaging used (GRAPPA, CAIPIRINHA, Multi-Band, Simultaneous Multislice)

As now indicated in table 2, the parallel imaging scheme used for diffusion-weighted imaging on 1.5T and 3T Siemens scanners was GRAPPA, for GE scanners was ASSET and for Philips scanners was SENSE.

- Regarding the single-rater reproducibility of tumor volume, which showed higher variability driven largely by two very small lesions, please clarify how this relates to the minimum 1 cm lesion size for the specified inclusion criteria. Are those exclusively post-treatment lesions that regressed? If so, would they not be the most important to capture ADC changes pertaining to tumor regression, since the lesions were the most responsive to treatment as measured by standard criteria (e.g., RECIST)?

These were the smallest tumours in the baseline reproducibility study at 1.3 and 1.26cm3. The distribution of tumour volumes was very skewed with most of the tumours being significantly larger than this.  Tumours for baseline reproducibility were deliberately chosen to represent the entire range of tumour size so that some tumours at the lower limit of acceptability would deliberately be included.

- In the discussion, I would consider the alternate interpretation that ADC-measurable tumor microenvironment changes do happen in response to treatment, but not necessarily at the pre-specified 14-day interval that was analyzed.

We strongly agree with the reviewer and have now added the text “Finally, we cannot exclude the alternate interpretation that ADC-measurable tumour microenvironment changes do happen in response to treatment, but not necessarily at the pre-specified 14-day interval that was analysed”.
